# Computational Thinking and Modeling: A Quasi-Experimental Study of Learning Transfer

Line Have Musaeus [1] and Peter Musaeus [2,*]

1    Center for Computational Thinking & Design, Aarhus University, DK-8200 Aarhus, Denmark; lh@cs.au.dk
2    Centre for Educational Development, Aarhus University, DK-8200 Aarhus, Denmark
*    Correspondence: petermus@au.dk

**Abstract:** This quasi-experimental study investigated the impact of computational learning activities on high school students' computational thinking (CT) and computational modeling (CM) skills. High school students (n = 90) aged 16 to 19 engaged in activities using computer models versus textbook-based models in mathematics and social science. The results indicated that students using computer models showed significant improvements in CT and CM skills compared to their peers in conventional learning settings. However, a potential ceiling effect in the CT assessments suggests that the test may not fully capture the extent of skill development. These findings highlight the importance of integrating computational learning activities in education, as they enhance students' abilities to apply these skills beyond the classroom.

**Keywords:** computational thinking; computational modeling; transfer of learning; computing education; high school education; Bebras test; NetLogo

## 1. Introduction

High school education aims to prepare students to tackle new problems. This is analogous to the phenomenon known as transfer of learning where knowledge, skills, and attitudes acquired in one context are applied in another. Much educational research has been undertaken to comprehend the transfer of learning [1,2]. Specifically, in the context of computing education, learning computing skills seems to prepare students for problem-solving across multiple disciplines [3].

This is particularly relevant when examining the relationship between computational thinking (CT) and computational modeling (CM). CT involves problem-solving processes that include abstraction, algorithm design, and data analysis, while CM refers to the creation and use of computational representations to simulate and study complex systems. Understanding how students can transfer their learning between CT and CM can provide insights into designing more effective educational strategies that facilitate deeper comprehension and broader application of computational skills.

Assessing the transfer of learning in computing education involves understanding how students apply their knowledge and skills across different contexts. This assessment provides insights into how students are equipped to apply their knowledge in diverse scenarios, thereby supporting the development of their thinking and the improvement of education [4]. The ability to generalize and apply computational skills across various domains enhances students' analytical and problem-solving abilities [5,6].

This article reports on a study of transfer of learning in high school education. High-school students are at a pivotal stage in their development where the ability to transfer learning can significantly influence their future. The objective of the study was to advance understanding of the assessment of CT and CM and to explore the transfer of learning CT. This has ramifications for assessment as well as how educators might prepare students for real-world challenges in future studies and work.

The research question was as follows: How do skills in Computational Thinking (CT) transfer to skills in Computational Modeling (CM)? This question was explored through an intervention study involving pre- and post-assessments of CT and CM skills among high school students in both an intervention and a comparison group. The findings aim to inspire further research in computing education and provide guidance to high school teachers and researchers on preparing students for educational and career demands that require computational skills. Based on the results, the study recommends integrating computational learning activities into existing high school subjects, such as mathematics and social sciences, to promote the transfer of students' CT skills to CM.

Before presenting the results, Section 2 introduces and defines CT and CM in the context of learning transfer. This section also discusses methods for measuring CT and CM. Following this, in Section 3, the methods used in the study are detailed, including pre- and post-measurements of students' CT and CM skills. In Section 4, the results are then presented, followed by a discussion in Section 5.

## 2. Background

CT refers to the thought processes in formulating and preparing a problem for a computational solution [7–9]. CT involves fundamental concepts such as problem-solving by decomposition, algorithmic and logical thinking, modeling, data, and pattern recognition, system thinking, and abstraction [3,10].

CM involves creating and using abstractions to simplify, represent, and analyze complex, dynamic, and emergent phenomena to predict outcomes and understand causal effects [11]. Research has demonstrated that CM helps students master different subjects, including computer science [12–14], physics [15], biology [16,17], and the liberal arts [18,19].

Various studies have investigated the validity of different tests and assessment methods relating to computing [20,21]. There are various methods for assessing CT and CM skills, such as student drawings [22], think-aloud protocols [23], peer reviews [24], performance-based tasks [25], portfolios [26], dynamic code analysis [27], concept inventories [28], scenario-based assessments [29], coding assignments [30], and rubric-based evaluations [31]. Understanding the core components of CT and CM is essential for effectively assessing these skill sets and teaching them in the high school curriculum.

Seminal research conducted by Selby and Woollard [32], Weintrop et al. [10], Barr and Stephenson [33], and Grover and Pea [34] provide the basis for assessing and enhancing students' abilities in these critical areas. Table 1 presents an overview of the key components of CT and CM.

**Table 1.** Components of CT and CM.

| Component | Definition |
|---|---|
| *Computational Thinking (CT)* | |
| Abstraction | Simplifying or hiding details to get at the essence of something of interest. |
| Decomposition | Breaking a problem into smaller parts that can then be solved separately. |
| Logical Thinking | Thinking clearly and precisely, including avoiding errors and attention to detail. |
| Algorithmic Thinking | Solving a problem in an efficient step-by-step manner, focusing on selection, sequencing, and iteration. |
| Evaluation | Examining a solution and judging whether it is doing what it is designed to do and how it could be improved. |
| Generalizations | Taking the solution, or parts of the solution, to a problem that may be reused and reapplied to similar or unique problems. |

**Table 1.** *Cont.*

| Component | Definition |
|---|---|
| *Computational Modeling (CM)* | |
| Model Creation | Designing, constructing, using, and assessing computational models to simulate real-world processes or phenomena. |
| Model Simulation | Running computational models to test hypotheses and predict outcomes. |
| Model Analysis | Interpreting the results of model simulations to draw conclusions or make predictions. |
| Model Validation | Comparing model predictions with real-world data to assess accuracy and reliability. |
| Model Refinement | Improving models based on validation results and new information. |

As shown in Table 1, CT and CM are two interrelated domains that offer a framework for understanding computational problem-solving in an educational context. CT focuses on the thought processes involved when students formulate problems and devise solutions that can be executed by a computer. It encompasses activities such as abstraction, decomposition, logical and algorithmic thinking, evaluation, and generalization. CM, on the other hand, applies these thinking skills to create and refine models that simulate real-world phenomena. This includes model creation, simulation, analysis, validation, and refinement. While CT is arguably a more foundational skill necessary for problem-solving, CM is a subset or applied skill used to construct and refine models. In summary, CT serves as a precursor, equipping students with the cognitive tools needed to think like a computer scientist, whereas CM leverages this to address real-world scenarios. Both are essential in a modern high school curriculum (in computing and other subjects), with research on CT offering the conceptual framework and CM arguably delivering more specific applications.

Transfer of learning is a well-researched phenomenon in psychology and educational science. Transfer of learning can be categorized into near and far. Near transfer of learning occurs when the contexts or learned material are similar, facilitating easier application of skills, such as applying math skills learned in class to solve similar problems in homework [2]. Far transfer of learning, on the other hand, involves applying skills to dissimilar and more abstract contexts, such as using problem-solving strategies learned in mathematics to address issues in social sciences or real-world situations [1].

Table 2 illustrates the types of tests used in the transfer of learning, memory, knowledge, and skills, and how the tests translate to the CT and CM contexts.

**Table 2.** Types of transfer of learning tests.

| Test Type | Description | Translated to CT and CM Context |
|---|---|---|
| Near Transfer Tests | Assess the ability to apply learned skills in similar contexts. Example: applying previously learned coding techniques to solve similar computational problems, such as debugging similar types of errors in the same programming environment [1]. | Students solve tasks that require them to apply previously learned programming skills to new but similar problems. Skills include debugging different segments of code that exhibit similar types of errors and applying algorithmic thinking to new but related contexts [34,35]. |
| Far Transfer Tests | Evaluate the ability to apply learned skills in different and more abstract contexts. Example: applying computational thinking skills learned in a programming class to solve real-world problems in different domains, such as using algorithmic thinking to optimize logistics or other processes [2]. | Students apply computational thinking skills acquired in a computer science class to tackle problems in different domains, such as optimizing a supply chain or creating a model for social behavior. This demonstrates the transfer of skills to varied subjects and real-world contexts [36]. |

Transfer of learning in the context of computing education can be evaluated through various theoretical frameworks and practical approaches. Guzdial and Nelson-Fromm [37] discuss the purpose-first theory of transfer, emphasizing that transfer occurs more readily at the functional level rather than the structural or behavioral level. They suggest that the motivation to learn programming plays a crucial role in facilitating transfer, especially for students studying computing as a general education subject rather than for professional development [37]. Guzdial and Nelson-Fromm used the structure–behavior–function (SBF) model to analyze transfer among students. They found that knowledge transfer is more likely to occur at the functional level, where students focus on the purpose and outcome of the code rather than its syntactic details. This aligns with the findings that students who are less motivated to learn programming for its own sake but are interested in its applications in other domains may benefit from a purpose-first approach to teaching programming [37].

Saba et al. [38] explored the relationship between CT, systems understanding, and knowledge transfer. Their study found that engaging students in constructing computational models significantly improved their conceptual understanding of science, systems thinking, and CT. They also showed that such activities promoted both near and far transfers, with a medium effect size for far transfer of learning, interpreted following Cohen's guidelines, where an effect size (d) below 0.2 is considered small, around 0.5 is medium, and 0.8 or above is large [39]. This suggests that integrating CT with domain-specific content can enhance the transfer of learning across different contexts [37].

Ye et al. [40] conducted a systematic review and meta-analysis on the transfer effects of CT across various subject areas beyond computer science. Their findings indicated that CT skills positively impact learning outcomes in mathematics, science, and engineering. The review study identified several instructional elements that promote transfer, including engaging students with new information, enabling them to demonstrate competence, and applying skills to real-world problems. These elements align with constructivist principles, emphasizing active learning and the application of knowledge in different contexts [37,40].

Based on the research described above, we chose to include pre- and post-assessments of students' CT skills and assessments of students CM skills continuously throughout the study to investigate learning transfer.

## 3. Methods

### 3.1. Study Design

The study was designed as a quasi-experimental intervention-control study employing pre- and post-intervention assessments to evaluate students' CT and CM skills in two groups of high school students: an intervention and a comparison group.

The chosen quasi-experimental design incorporates a passive control group to enhance the evaluation of different instructional approaches, specifically comparing conventional methods versus computational learning activities. Conventional methods refer to standard instruction using established textbooks, teacher-led lectures, and discussions to explore mathematical and social science phenomena, with the same duration of teaching hours as the intervention group.

This design facilitates a direct comparison of students' CT and CM skills between those exposed to innovative teaching methods and those following existing approaches. In this setup, the passive control group, which continues with conventional teaching methods, serves as a baseline. This comparison helps establish the extent to which existing education fosters the development of CT and CM skills, which are increasingly crucial in both academic and everyday contexts.

In the fields of mathematics and social sciences, models are fundamental tools for understanding key phenomena. Students typically encounter these models through textbooks. This study aimed to investigate how well this existing approach prepares students for the computational models they are likely to encounter in their everyday lives, identifying any potential gaps in current educational practices. Additionally, the study sought to determine

whether integrating CT and CM skills into mathematics and social sciences education better equips students to engage with computational models in real-world scenarios.

The quasi-experimental design, with its passive control group, is well-suited for this investigation as it allows researchers to measure the impact of computational learning activities on students' competencies, highlighting any significant improvements compared to conventional methods. This design is both practical and ethically sound, as it does not withhold educational opportunities from any group but instead leverages existing classroom practices to evaluate the effectiveness of new teaching methods. By comparing results between the experimental group and the passive control group, the study design can provide evidence of the potential benefits of integrating CT and CM skills into the high-school curricula, ultimately enhancing educational outcomes and better preparing students for the challenges they will face in the 21st century.

The interventions are described in detail in Section 3.3. Students in the intervention group participated in both a math intervention and a social science intervention, one to two weeks apart, as shown in Figure 1.

* Descriptions of assessment items CT1–CT5 and CM1–CM6 can be found in Tables 5 and 6.

**Figure 1.** The study design.

Both interventions included computational learning activities involving computer models of domain-specific phenomena. As shown in Figure 1, students in the comparison group participated in interventions that followed conventional teaching methods. The conventional teaching class served as a control group to establish a baseline for comparison. By contrasting the outcomes of students engaged in existing instructional methods with those participating in computational learning activities, we aimed to evaluate the added value and effectiveness of the new instructional approaches. This quasi-experimental design ensures that any observed differences in CT and CM skills are attributable to the intervention itself, rather than to variations in teaching methods, as outlined by Campbell and Stanley [41]. Consequently, the conventional teaching control group provides a robust context for assessing the true impact of computational learning interventions.

Students' CT and CM skills were assessed before, immediately after, and 3–4 weeks following the interventions (see Figure 1). There is likely convergent validity between the tests for CT and CM, given the close relationship between these two phenomena. The learning activities and models used in this study can be found on this website: https://graspit.dk/.

### 3.2. Participants

The study included 90 Danish high school students (16–19 years old, average age 17.8) from four schools, with a gender distribution of 40% female and 60% male. Over 75% had no programming experience, and only 1% had extensive experience. None had prior computing education.

Students were divided into intervention and comparison groups based on their classes, which were randomly assigned by the school. The intervention group participated in CT learning activities of a 120 min duration in mathematics and in social science, guided by two key principles.

First, the Use–Modify–Create Principle [42] involved students initially using a computer model to learn about a phenomenon. They then modified the code to improve the model and finally created new procedures to further enhance the model.

Second, the CMC Approach [43] integrated coding activities with modeling the phenomenon and connected these activities to the students' existing knowledge of the subject matter.

In the intervention group, students first tinkered with the models' interfaces and then modified the code to better represent content within mathematics or social science. This constructivist approach allowed students to experiment with the models before being assigned specific tasks.

Students in the comparison group received conventional instruction, which was of equal duration (120 min each session) and covered the same mathematical and social science concepts, utilizing textbooks. Table 3 presents an overview of the two groups and their respective teaching methods.

**Table 3.** Overview of study groups.

| Group | Teaching Method |
| --- | --- |
| Intervention group | Two teaching lessons (conducted by teachers, 120 min each). Learning activities with NetLogo 6.1 models, modifying code, changing variables, loops, and introduce new procedures. |
| Comparison group | Two conventional teaching lessons (conducted by teachers, 120 min each) using textbook models, answering subject-related questions. |

Students' content knowledge in mathematics and social science was evaluated by the teachers after the interventions and reported to researchers. Both groups participated in pre-intervention (Test 0), immediate post-intervention (Tests 1 and 2), and follow-up assessments (Test 3) administered 3–4 weeks after the interventions (see Figure 1). Figure 2 below shows student participants working individually with computer models in mathematics. Apart from the students, eight teachers (two female, six male) participated.

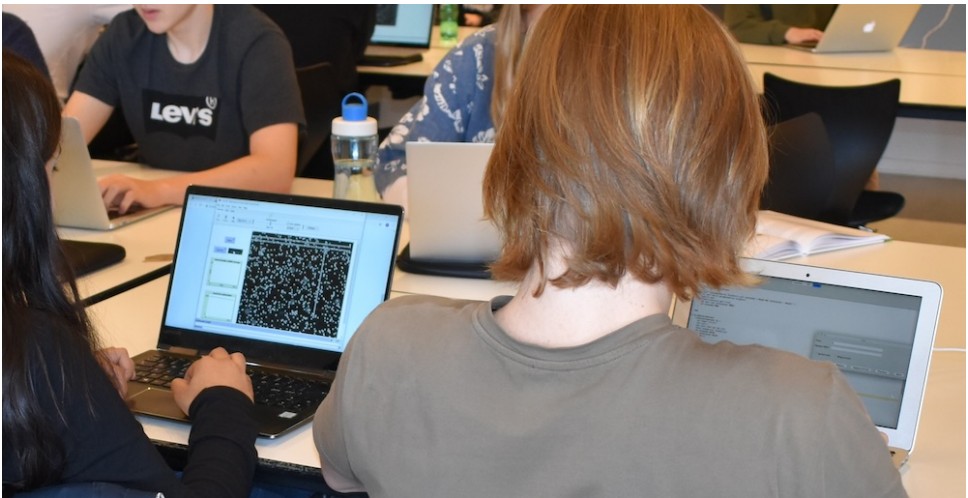

**Figure 2.** High school students working with computer models in the classroom.

Informed consent was obtained from all participants, and the study was exempt from Aarhus University Ethical Committee approval due to its non-invasive nature. Participants were assured of anonymity and informed about the research purpose, data usage, and the absence of risks. The study adhered to all local and national research laws, with rigorous data management protocols. Written consent for the use of pictures was obtained from all participants.

### 3.3. Interventions

The interventions in this study for both groups lasted 120 min, involving students, in the intervention group, interacting with and modifying computer models of mathematical or social science phenomena designed in NetLogo, an agent-based modeling environment [44]. During these sessions, students engaged in active exploration, experimentation, and problem-solving requiring CT [45,46]. These interventions were conducted by the students' regular teachers in math and social science, respectively.

The interventions combined activities within the concept of CT with content in mathematics or social science, focusing on CM. The goal was for students to engage, interact, and communicate with each other on both subject matter and computing activities, mediated by the computer. This approach was inspired by Seymour Papert's constructionist perspective, where students actively construct their own knowledge and skills [45]. The approach is illustrated in Figure 3.

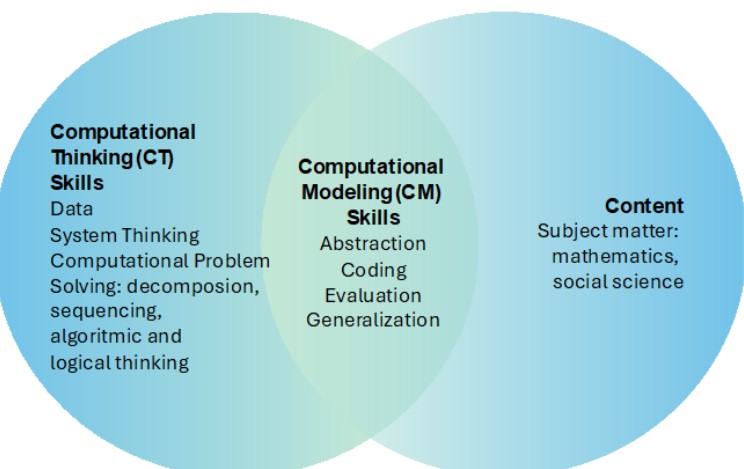

**Figure 3.** Interventions on CM, CT, and subject matter.

Material

All students, in both the intervention and comparison groups, were taught the SIR model in mathematics, a compartmental model that describes the development of diseases through coupled differential equations. The SIR model is a common topic and one of the learning objectives of high school mathematics education at the highest level. Typically, this topic is introduced using textbook materials, with students in the comparison group solving a set of exercises using spreadsheets.

In contrast, students in the intervention group were introduced to a NetLogo model simulating disease development with various adjustable variables. These students used the computer model to solve a similar set of exercises by modifying the code within the model and simulating how a disease develops.

In social science, both groups were introduced to the same topic "The economic cycle", a central learning objective in this field. The intervention group again used a NetLogo model to simulate the economic cycle, adjusting variables by modifying the code to solve a set of problems related to the values of these variables. The comparison group, on the other hand, learned the topic through textbook materials, models, and exercises that addressed the same problems as those posed to the intervention group.

### 3.4. Teaching the Teachers

Mathematics and social science teachers participated in a professional development (PD) course for STEM and social science education, led by the first author. Six out of eight teachers in the intervention group developed their own computer models, while the remaining two taught the comparison group and did not participate in the PD course.

During two seminar days, teachers were introduced to the NetLogo programming environment and paired up to help each other prepare the interventions. They designed their computer models and learning activities with assistance from teaching assistants. Subsequently, the interventions were conducted by the teachers using their developed activities without the presence of researchers. Students engaged with the models and modified the code to solve tasks.

*3.5. Measuring Instruments*

For assessment, we used selected questions from the Bebras test and a computational modeling test in the form of NetLogo programming tasks [36,47]. The Bebras test assesses CT skills through context-rich problems, while NetLogo tasks assess students' ability to construct, implement, and evaluate computational models. This dual approach aimed to measure students' CT and CM skills in different contexts.

Pre- and post-intervention assessments included quantitative data (multiple-choice questions) and qualitative data (open-ended questions). Together with Figure 1 shown in Section 3.1, Table 4 outlines our assessment approach.

**Table 4.** Assessment approach with two assessment methods.

| Assessment Method | Description | CT Skills Assessed | CM Skills Assessed | Context |
|---|---|---|---|---|
| Adapted Bebras Test, used before and after the interventions in test 0 and test 3 (See Figure 1 in Section 3.1 Study design) | Uses modified Bebras questions to assess CT skills such as algorithmic thinking, pattern recognition, and debugging [48,49]. | Algorithmic thinking, pattern recognition, debugging | - | Computer-based, problem-solving tasks |
| NetLogo Programming Tasks, used in test 0, 1, 2, and 3 (See Figure 1 in Section 3.1 Study design) | Students use and modify models in the NetLogo environment to explain and solve problems and demonstrate understanding of computational concepts and models [27,36,45]. | Abstraction, decomposition, algorithmic thinking | Use and modification of model creation, simulation, abstraction | Simulation and modeling tasks in NetLogo |

To investigate the gains in students' CT skills, we used pre- and post-intervention assessments with multiple-choice questions from the Bebras CT test [47,48], which has previously been applied to high school students by Lee et al. in Taiwan [50]. The CT test questions focused on algorithmic and logical thinking, including the ability to decompose and pattern recognition. These cognitive skills have been widely identified by researchers as important components of CT [51,52].

Five questions from the Bebras test [35] were selected and used for testing students' CT skills in this study. The five questions were selected based on what the researchers and one former high school teacher perceived as easy (CT1, CT2, and CT5) and difficult (CT3 and CT4) questions. These questions were chosen because they focus on algorithmic thinking and visual as well as conceptual-based thinking [48,49]. The wording of the questions is shown in Table 5.

To evaluate the transfer of learning between students' CT and CM skills, we also included a CM test in the study. The CM test, developed for a similar age group in STEM classrooms [36,53], included open-ended questions as shown in Table 6.

**Table 5.** CT test with five questions.

| Multiple Choice Question | Wording |
|---|---|
| CT1 | A chemistry teacher puts five bottles on a table. He places them so that each bottle is visible. He places the first bottle at the back of the table, then places each new bottle in front of those already placed on the table. What is the correct order of bottles from first to last? |
| CT2 | Help the green robot to get out of the maze by using a sequence of the proposed movements. |
| CT3 | Kasper is having a party and has a roll of colored paper he wants to hang up as a decoration. The paper has colored squares in three different colors (yellow, red, blue) in a repeating pattern. Kasper's friend, Mathias, has torn out some of the paper, as can be seen in the diagram below. Mathias says that he will give Kasper the missing piece of paper back if he can work out how many-colored squares have been torn out. How many squares are missing? |
| CT4 | Kamilla has discovered five different magic potions for cats:<br>One potion makes the ears of cats grow longer.<br>The second potion makes the teeth of cats grow.<br>A third potion curls the whiskers.<br>A fourth potion colors the cats' noses white.<br>The fifth potion changes the color of the eyes to white.<br>Kamilla pours each of the potions into a separate mug and pours clean water into her personal mug so there are now six mugs in total. The mugs are labeled A, B, C, D, E, F. Unfortunately, Kamilla has forgotten to note which mug contains which drink. Can you help her? |
| CT5 | The agents Billy and Berta write secret messages to each other. Billy would like to send Berta the following secret message: MØDAGENTENBILLYKL6 He writes each character in a table with 4 columns from left to right and row by row starting from the top. He puts an X in the fields that are not used. The result can be seen below.<br>Berta used the same method to write back to Billy. The secret message she sends him is:<br>OGE!KMRXJØOXEDPX<br>What message does Berta send back? |

**Table 6.** CM test with six questions.

| Open-Ended Question | Wording |
|---|---|
| CM1 (open question) | Start the model and let it run for 1000 steps. Describe what happens to the number of strawberry pickers and strawberries during the 1000 steps: |
| CM2 (open question) | Describe the relationship between the number of strawberries and the number of strawberry pickers: |
| CM3 (open question) | Start the model again. This time try pressing the 'Frost' button while the model is running.<br>Describe what happens to the number of strawberry pickers when the frost destroys half of the strawberries. |
| CM4 (open question) | What do you think would happen to the number of strawberry pickers if frost destroyed 90% of all the strawberries instead of 50%? |
| CM5 (open question) | Write instructions that could be followed by a computer to simulate how birds can remove some of the strawberries in the model. |
| CM6 (open question) | All computational models are only approximations of reality. What are some ways in which this model is different from reality? |

The test was translated using the back-translation method and piloted with four students before administration. The test addressed topics unrelated to the phenomena and models taught in the interventions. As already shown in Figure 1, the tests were administered before the interventions (test 0), immediately after each of the interventions (test 1 and test 2), and 3–4 weeks after the interventions (test 3). Specific questions relating to real-world phenomena and the ability to problem-solve with a computer were included in the pre- and post-intervention assessments. The CM5 question was included to investigate students' ability to transfer CT skills to similar CM problems, and the CM6 question was included to investigate students' transfer of CT skills to real-world settings.

*3.6. Data Analysis*

Data analysis was conducted using Microsoft Excel. Quantitative data from CT test scores and qualitative data from student answers to open-ended questions were analyzed to determine the effectiveness of the interventions and the near transfer of learning. All data were found to be normally distributed by assessing the values of skewness and kurtosis as suggested by Holmes et al. [54]. Statistical methods, including two-factor ANOVA, and paired *t*-tests with measured effect size, Cohen's d [39], as elaborated by Haden [55], were used to evaluate improvements in CT and CM skills. A Cronbach's Alpha value of 0.973 was estimated, indicating high internal consistency of the survey, as interpreted by Brown [56] and Tavakol and Dennick [57].

For details about the CM questions, we refer to Table 6. For the multiple-choice questions in the CT tests, each answer was scored as either 1 for a correct response or 0 for an incorrect one. The open-ended questions in the CM tests were independently coded by two researchers, with scores assigned as 2, 1, or 0 based on the accuracy of the answers. A score of 2 was awarded if the response included all aspects of a correct answer, while a score of 1 was given for partially correct answers. A score of 0 was assigned when the response lacked any correct elements. For example, in response to the question, "Describe the relationship between the number of strawberries and the number of strawberry pickers", a score of 1 was given for an answer like "When there are many strawberry pickers, there are fewer strawberries", as it only partially addressed the relationship. A complete answer would need to consider the impact of the number of strawberries on the number of pickers over time. For other questions, such as CM6, a maximum score of 2 was given for answers that included all relevant simplifications and model elements. Similarly, in question CM5, a score of 2 was awarded for a comprehensive sequence of instructions for removing strawberries, with partial instructions receiving a score of 1. The mean score for each question within each group was then calculated.

## 4. Results

*4.1. Students' CT Skills*

The results obtained from administering the CT test before and 3–4 weeks after the interventions revealed a statistically significant variance in means within both the intervention and comparison groups, as determined by a two-factor ANOVA [$F(3, 232) = 6.043$, $p = 5.593 \times 10^{-4}$ for the intervention group and $F(3, 112) = 7.932$, $p = 8.483 \times 10^{-5}$ for the comparison group].

In the intervention group, post-intervention assessments revealed higher mean scores for most questions (CT1, CT2, CT3, and CT5). This contrasted with lower scores observed in the comparison group for questions CT2, CT3, CT4, and CT5, indicating students in the intervention group improved their CT skills.

Figure 4 illustrates the mean scores for each of the five CT-questions.

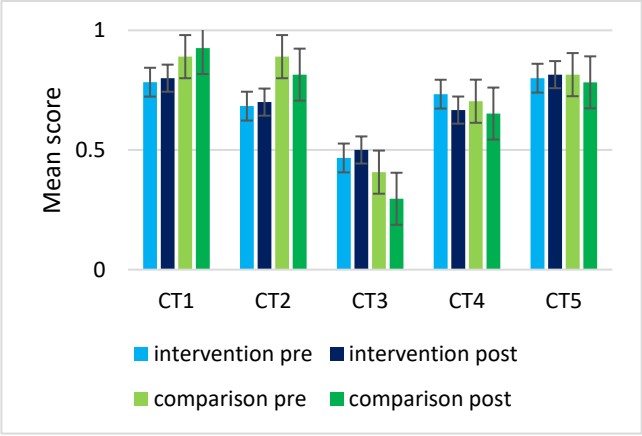

**Figure 4.** Mean scores in each of the five CT questions (CT1, CT2, CT3, CT4, CT5) in the CT test.

Both groups demonstrated proficiency in selecting correct answers for the multiple-choice questions, especially in the three questions (CT1, CT2, and CT5) categorized as easy by the researchers (see Section 3.5 Assessment). The high scores in general could indicate a "ceiling effect" in which there is an upper limit on responses in a survey or questionnaire, and a large percentage of respondents score near this upper limit [58]. The ceiling effect could have produced false-positive outcomes and might have deflated the effect in the questions.

### 4.2. Students' CM Skills

When investigating students' CM skills, there was no statistically significant difference in the scores between the intervention and comparison groups across the first three tests. The results of the *t*-tests were as follows: Test 1 (t(358) = 1.648, *p* = 0.460, d = 0.11), Test 2 (t(358) = 1.648, *p* = 0.075, d = 0.13), and Test 3 (t(358) = 1.649, *p* = 0.163, d = 0.14). However, three to four weeks after the interventions, the test scores differed statistically significantly between the intervention and comparison groups, indicating a retention of CM skills learned by students in the intervention group. The results showed a medium effect size (t(358) = 1.649, *p* = 0.0125, d = 0.29).

Figure 5 depicts the development of students' CM skills.

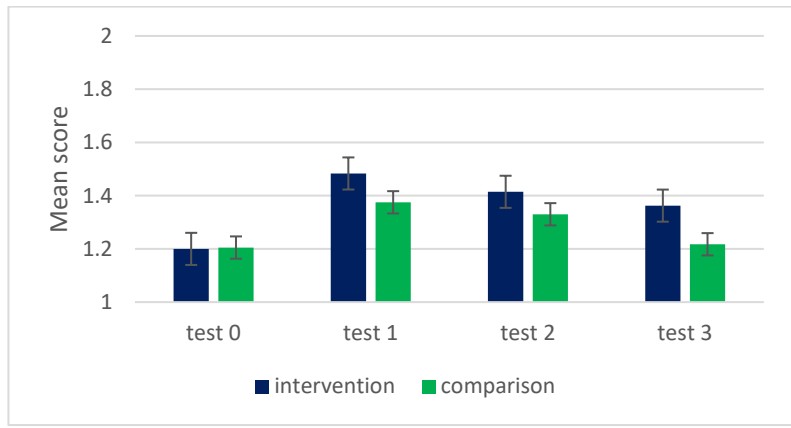

**Figure 5.** Development of students' CM skills: Mean scores of students' answers to all questions in the CM test.

Overall, students in the intervention group demonstrated a higher frequency of correct responses compared to the comparison group (see Figure 5). The intervention group exhibited a 14% increase in CM skills from test 0 to test 3, whereas the comparison group showed only a 1% improvement.

### 4.3. Students' Transfer of CT and CM Skills

4.3.1. Comparing a Computer Model to a Real-World Phenomenon

Students from both groups were given an open-ended question (see CM6 in Table 6) as part of test 0 and test 3. This question addressed two aspects: reflections on how a specific computer model represented a real-world phenomenon and any simplifications included in the model. The question, which was new to all students, was as follows: "All computational models are only approximations of reality. What are some ways in which this model is different from reality?". Figure 6 shows students' answers when comparing computer models to real-life phenomena (maximum score 2).

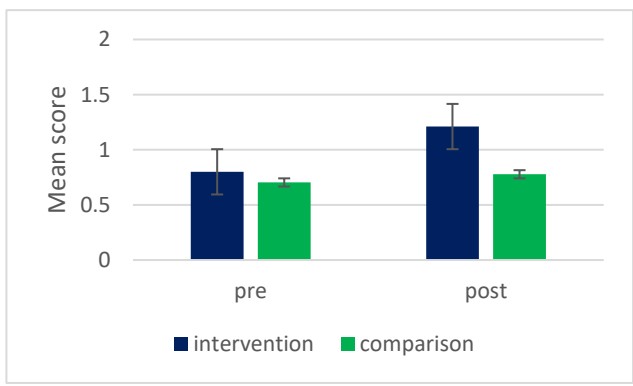

**Figure 6.** Mean scores of students' answers pre- and post-interventions.

The students in the intervention group showed statistically significant improvement in their ability to identify elements and simplifications in the model related to the real-life phenomenon from pre- to post-intervention (t(118) = 1.658, *p* = 0.0004, d = 0.66), indicating a medium effect size, d. In contrast, the students in the comparison group did not show significant improvement (t(58) = 1.675, *p* = 0.3517, d = 0.71). Furthermore, when comparing the answers from the two groups after both interventions (test 3), there was a statistically significant difference in the mean scores between the two groups, with the intervention group performing better than the comparison group (t(88) = 1.663, *p* = 0.0015, d = 0.64), indicating a medium effect size. This difference was not observed before the interventions (test 0), suggesting that students from the intervention group gained expanded CT skills during the interventions compared to the comparison group, which they could transfer to their answers in the CM test.

4.3.2. Communication with a Computer

Before the interventions and 3–4 weeks after (during CM tests 0 and 3), all students were asked to describe a procedure as a sequence of instructions related to a problem within a specific computer model as part of the CM test. This task, like question CM6, aimed to assess students' expanded CT skills. The question was: "Write instructions that could be followed by a computer to simulate how birds can remove some of the strawberries in the model" (see CM5 in Table 6). This question was new to all students in both groups.

Figure 7 shows scores of students' instruction sequences before and after the interventions.

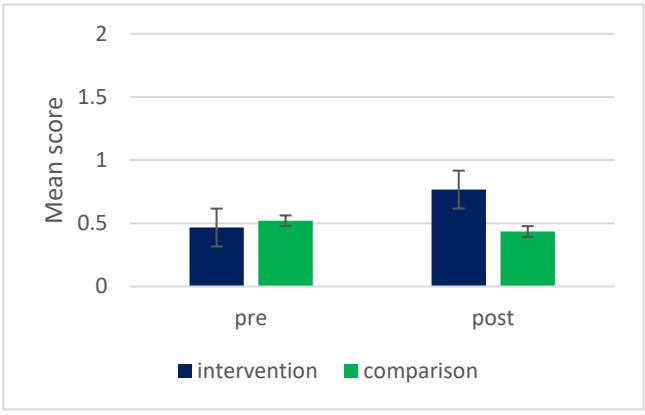

**Figure 7.** Mean scores of students' instruction sequence descriptions: pre- and post-interventions.

Figure 7 shows that no statistically significant difference existed between the two student groups before the interventions (t(88) = 1.659, *p* = 0.2908, d = 0.61). However, after the interventions, students in the intervention group demonstrated statistically significant improvement in describing sequences of instructions compared to the students in the

comparison group, with a large effect size (t(88) = 1.664, *p* = 0.0177, d = 3.43). This indicates that the intervention group gained expanded CT skills through their participation in computational modeling activities and were able to transfer these skills.

Students' answers to the CM test question described above (see CM5 in Table 6) were examined further and assessed for inclusion of loops, sequences, and a specific programming syntax in the answers.

Figure 8 shows the percentage of students using loops, NetLogo syntax, and problem-solving strategies in their instruction sequences, both before and after the learning activities.

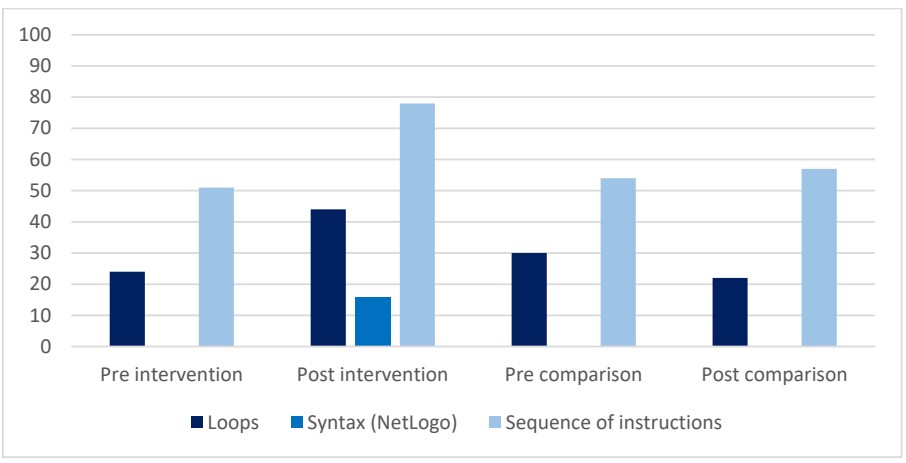

**Figure 8.** The percentage of students using loops, NetLogo syntax, and problem solving by describing a sequence of instructions before and after the learning activities.

As shown in Figure 8, the number of students able to describe the instructions using loops and sequences was low for both groups before the interventions (30% and 24%). After the interventions, a higher percentage (44%) of the students from the intervention group used loops and sequences in their descriptions, compared to the comparison group of students (22%).

Around half of all students (51% and 54%, respectively, see Figure 8), in both groups, were able to solve the problem by describing a sequence of instructions before the interventions. After the interventions, more than three-quarters of all students (78%) in the intervention group were able to describe a sequence of instructions for a computer to carry out, while approximately half of the students (57%) in the comparison group were able to do the same. Hence, the results presented in Figure 8 indicate that students gained CT skills from participating in the interventions and were able to transfer them from the CM learning activities.

A subgroup of students (16%) from the intervention group used specific NetLogo syntax in their answers after having participated in the interventions, although they were not asked to do so. No students from the comparison group used a specific syntax in answering the question.

In summary, students who used computational models and participated in computational learning activities during the interventions performed significantly better in both evaluating a computer model in relation to a real-world phenomenon and writing instructions that can be followed by a computer compared to students who used a textbook model. Overall, the intervention group showed a significant increase in their ability to understand and explain the modeled phenomenon. After the interventions, teachers evaluated and reported on students' content knowledge in mathematics and social science. They estimated that the learning gains among students in the intervention group were equal to those achieved through existing teaching.

## 5. Discussion

Transfer of learning is essential for problem-solving. Ensuring that students can transfer CT skills to different contexts prepares them to tackle real-world challenges and pursue advanced studies [49]. Assessing the transfer of learning effectively requires a combination of qualitative and quantitative methods. As demonstrated in this study, tools such as adapted Bebras test questions and NetLogo programming tasks offer a comprehensive evaluation of students' abilities.

The findings from this study suggest that targeted computational interventions for high school students, particularly those with little prior knowledge of computing (see Section 3.2, Participants), can significantly enhance their CT and CM skills. Specifically, the integration of computational activities in teaching mathematics and social science enabled students to develop CT and CM.

The effect for transfer of learning was robust when examining students' expanded CT skills (see Figures 6 and 7). The post-test administered up to four weeks after the last intervention (see Figure 1) assessed retention and indicated sustained learning. Moreover, students in the intervention group demonstrated an increased awareness of how computer models can represent real-world phenomena. Notably, more students were willing and able to communicate with computers by framing problems as a sequence of instructions to a computer after participating in CM-related learning activities (see Figure 8). This task, as highlighted by Curzon et al. [59] and Weintrop et al. [10], can be classified as CT, supporting the integration of CM in various high school subjects, such as mathematics and social science. This approach could engage a diverse group of students, including those who are either more or less motivated in computing.

An analysis of students' initial CT skills revealed that most students were already scoring high, indicating a possible ceiling effect. A ceiling effect occurs when a large proportion of participants achieve high pre-test scores, limiting the observable gains during the intervention [58]. This phenomenon is common in studies involving students with a strong interest in and knowledge of the subject. In this study, students from both groups had elected to study mathematics at the highest level, suggesting a pre-existing proficiency in logical and algorithmic thinking—the primary focus of the CT test.

Consequently, most students achieved near-perfect scores. Future research should ensure that the Bebras problems used are sufficiently challenging or consider alternative CT tests that allow for a broader distribution of responses.

Another challenge in traditional pre-post designs is the response shift bias, which can lead to inaccurate pre-test ratings. This bias occurs when students overestimate their knowledge and abilities at the start of a course or intervention [60]. Post-test scores, therefore, often provide more accurate assessments as students gain a better understanding of the questions or benchmark themselves against their peers. This response shift might inflate the perceived effect of the intervention [61]. To mitigate this bias, this study used multiple methods to assess students' CT skills, including Bebras questions and tasks such as describing sequences of instructions for a computer to execute (see Figures 7 and 8).

Perkins and Salomon [62] describe low road transfer as the automatic triggering of well-practiced routines in similar contexts, relying on surface characteristics. In contrast, high road transfer requires mindful abstraction and deliberate effort to apply learned principles to new and different contexts. Perkins advocates for a whole approach to teaching, emphasizing the integration of authentic experiences and reflection to foster both types of transfer [63].

In our study, it is possible that students improved their CM skills simply through participation, leading to better scores over time. Analysis of the CM test scores revealed that students in both groups performed better on subsequent tests, potentially indicating a low road transfer, where test conditions activated well-practiced routines like those in the learning context. However, the final test (Test 3) showed a statistically significant difference in scores, with the intervention group outperforming the comparison group. This suggests that the intervention effectively facilitated the learning and retention of CM skills.

### 5.1. Future Directions

This study aimed to measure students' ability to apply CT and CM skills in both theoretical and practical contexts closely related to their initial learning experiences. Future research could explore far transfer, which involves applying these skills in more distant and unrelated contexts. For example, far transfer might include learning algorithmic thinking in biology and then using that skill to optimize supply chain logistics. Similarly, it could involve assessing the impact of learning CT in computing on students' critical thinking in philosophy, or examining how CT enhances creativity in social sciences, ultimately helping students solve real-world problems. However, measuring far transfer is often more challenging than near transfer and may require longitudinal research designs, which are beyond the scope of this study.

As we continue our research on the transfer of CT and CM skills in high school education, future studies could focus on the underlying mechanisms that lead to either high or low road transfer. CT and CM are not unidimensional constructs and do not function as entirely separate processes.

Our goal is to understand how CM can enhance both students' CT and their comprehension of real-world phenomena. Far transfer tests evaluate the ability of students to apply learned skills in different and more abstract contexts—such as applying CT skills acquired in a programming class to solve problems in various domains, like optimizing logistics or other processes. This exemplifies high road transfer, where skills learned in one context are applied to diverse subjects and real-world challenges. In this study, we focused on the near transfer of learning between CT and CM rather than the far or high road transfer (see Table 2).

### 5.2. Limitations

The design of the learning activity may have limited the students' ability to fully express their CT and CM skills. The concept of generalization, which is known to be particularly challenging to master [48,64], was not clearly observed, as students lacked the opportunity to apply parts of their code to different models. Moreover, certain elements of the assessment methods, including visual and written expressions, did not adequately support the systematic multimodal representations that are integral to computational modeling tasks (e.g., shapes and arrows). Previous research has indicated that the ability to express one's ideas can be both supported and constrained by the choice of medium and the nature of a task [64,65]. Future research on students' transfer of learning between CM and CT should consider incorporating broader assessment methods. These methods should involve procedural, semantic, visual, multimodal, and embodied forms such as gestures [66,67] to better capture the range of students' computational skills and understanding.

To ensure effective transfer of learning, high school teachers should incorporate CT and CM in social science, mathematics, and other subjects. Integrating CT and CM can help students apply their understanding in diverse contexts, enhancing their ability to solve complex problems. This approach not only builds students' confidence and sense of accomplishment, but also fosters a deeper understanding of the subject matter [3,68]. Furthermore, as mentioned in the introduction, learning theorists as David Perkins and Jeanne Ormrod have asserted that teaching principles to students seem more easily to lead to transfer of learning than teaching discrete facts [63,69]. Therefore, to enhance students' self-efficacy [68], it is important to incorporate activities that improve their understanding of the principles underlying CT and CM. Assessments should evaluate how students engage in logical and algorithmic thinking using these principles, ensuring they can apply their knowledge effectively across different subjects.

Integrating principles of CT and CM into high school education might enhance students' transfer of learning and lead to expanded CT skills. By focusing on teaching the principles and encouraging practical application, teachers can help students develop the computational skills and confidence needed to succeed in various challenging tasks beyond the classroom.

Teaching CM can enhance students' initial understanding of CT, particularly in areas like abstraction and decomposition. For example, in social science, students can simplify the understanding of urban development by focusing on key factors such as housing and breaking down social segregation into causes and effects. In math, they can abstract population growth to essential elements like birth rates and decompose it into growth rate calculations and factor analysis. In physics, they can simplify projectile motion by focusing on gravity and initial velocity and breaking it down into horizontal and vertical components.

Through CM, the initial understanding of elements of CT might lead to a fuller, expanded understanding of CT and the social science curriculum or social phenomena in question. For instance, integrating CT and CM in social science can help students model and analyze complex issues by teaching them decomposition. Students can decompose social segregation by examining its causes (such as economic disparity, housing policies, and educational inequality) and effects (like community division and unequal opportunities). In urban development, they can break down the topic into components such as infrastructure, transportation, housing, and economic factors. For social unrest, they can analyze contributing factors (political instability, economic hardship, and social injustice) and resultant impacts (protests, policy changes, and societal disruption).

Through computational modeling, students learn to break down problems into manageable parts, facilitating a deeper understanding of each element's role and interaction. In mathematics, they identify patterns and apply CM to real-world problems. In natural science, CT principles guide experimental design, while CM helps students visualize and model natural phenomena.

## 6. Conclusions

This study highlighted the importance of transfer of learning in high school education and demonstrates the effectiveness of targeted interventions using NetLogo to enhance students' CT and CM skills. Further research is needed to gather and triangulate a broader range of evidence and assessment strategies on the transfer of learning between CT and CM. Implementing robust assessment strategies and targeted interventions in high school education can enhance students' analytical skills and their understanding of scientific methodologies through computational thinking and computational modeling. This approach supports the development of flexible and adaptive students who become prepared for advanced studies, for interesting careers, and to participate as citizens in a complex society.

**Author Contributions:** Conceptualization, P.M.; Methodology, L.H.M.; Software, L.H.M.; Validation, L.H.M.; Formal analysis, L.H.M.; Investigation, L.H.M.; Resources, L.H.M.; Data curation, L.H.M.; Writing—original draft, L.H.M. and P.M.; Writing—review & editing, L.H.M. and P.M. All authors have read and agreed to the published version of the manuscript.

**Funding:** Research was funded by the Velux Foundations. Grant no. 24635.

**Institutional Review Board Statement:** The study was exempt from approval by the Aarhus University Ethical Committee due to its non-invasive nature. Participants were assured of anonymity, informed of the research purpose, data storage and usage, and the absence of risks. The study complied with all local and national research laws, and data management protocols were rigorously followed.

**Informed Consent Statement:** Informed consent was obtained from all subjects involved in the study.

**Data Availability Statement:** The data presented in this study are available on request from the corresponding author due to legal and ethical reasons.

**Acknowledgments:** We wish to thank the students and teachers who participated in this study and our colleagues at Center for Computational Thinking & Design and Centre for Educational Development at Aarhus University for fruitful discussions on assessing the learning of complex computational skills.

**Conflicts of Interest:** The authors declare no conflicts of interest.

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
