# Peer review of "Computational Thinking and Modeling: A Quasi-Experimental Study of Learning Transfer"

_education, doi:10.3390/educsci14090980_

Round 1

Reviewer 1 Report

Comments and Suggestions for Authors

This paper describes a didactic intervention that studies its effects on Computational Thinking (CT), Computational Modeling (CM), and the potential of Transfer of Learning. As the development of such skills in High School is very important, I would like to thank the authors for their interesting ideas and the hard work they performed. The paper must be more precise and better structured to help readers understand their work.

The authors begin their introduction by presenting the innovation of their research. However, they use terms that need to be first defined and explained. It is pretty odd to give the research gap in the third line of the text. Introduction is an important section that describes the context of a study, the need to perform it, and how the goal will be achieved. I strongly recommend the authors define and describe their concepts before presenting the research gap. In the same section, it is also crucial to describe how you will answer the research question by describing what you will do. Also, please explain to whom the study’s results will be useful.

Then, the authors use an Assessment of CT and CM section to present some of the components of CT and CM. It is important to give a short description before Tables. Maybe it would also be interesting to add something about the connection between CT and CM, such as similarities, differences, etc. For example, both concepts apply coding to teach CT skills. On the other hand, I cannot understand why the following table is in this position as it has nothing to do with CT or CM, or the authors do not explain it… Since the authors have a third-section Assessment, could they include this Table there? The assessment section needs to be extended.   For example, the authors do not make clear whether the tests are used for the CT and CM skills assessment or both. Please include all the tests, including your choices, and then explain why the two tests were chosen. Maybe the authors could move the content of these two sections to the introduction section, except for the reason they used the Bebras test questions and NetLogo tasks, which can be moved to the methods in an instruments subsection. Otherwise, the authors need to extend the content of these two sections.

Methods

It is pretty hard for me to understand the authors' intervention well. Therefore, I will suggest the authors to use

-          A research setting section to only give a short description of their study, their study’s purpose, the study’s sample, and measures, e.g., CT, CM, ToL

-          A participants section to provide information, such as males and females, age per group, and how they will be divided into groups. Was the sample randomly distributed?

-          A materials section will give info about the curriculum, student projects, etc.

-          A measuring instruments section to provide information on the tests you used

-          A procedure section

-          A data analysis section: please be more specific about your measurements. Regarding qualitative data, you should also provide more information on what you sought in each answer. Please use Tables to be more clear.

Results are used to provide the results of a study. Statistical tests, such as two-factor ANOVA, have a particular way of presentation, using the defined variables and also F and p. The qualitative results are also connected to variables that have been previously explained (measuring instruments). The teachers’ comments and the researchers' observations are also part of the research's qualitative data. Consider including a coding schema if observations are also included.

I may have misunderstood some parts of your intervention, but it is not only my mistake. Please keep in mind that readers are going to read the paper if it is well structured while the instruments, measures, and results are clearly presented.

I did not spend a lot of time reviewing the discussion and conclusions sections as I am not quite sure about what you are measuring and how. Nevertheless, I will comment on the abstract section, which is too long and includes unnecessary information. An abstract needs consideration as it could convince somebody to read a paper. An abstract starts with the context and continues with the research gap (why this research must be done). Then, the method must be described (e.g., a between experiment, etc). Some important results, positive and negative, follow, and in the end, the study’s implications. Each of the above must be no more than one sentence.    

Author Response

"Please see the attachment".

The reviewer rightly remarked: "...I would like to thank the authors for their interesting ideas and the hard work they performed. The paper must be more precise and better structured to help readers understand their work." A big thanks to this reviewer for the detailed and relevant criticism of the paper that has led us to a comprehensive rewrite - all, very insightful reviewer remarks, every remark was heeded to as explained in pdf and can be seen in document with track changes.

Reviewer 2 Report

Comments and Suggestions for Authors

The article is very interesting and well written and understandable. However, there are a few suggested changes that should be considered:
- It is a quantitative study, but the hypotheses as well as the hypothesis tests are missing.
- I would reconsider Figure 3, even if the eyes are blinded, it is rather unusual and also ethically questionable at least to present a photo of the participants in an anonymous participation.
- It would be important to include a few more recent references. There are only a handful of sources that are three years old, some sources are even from the last century.

Comments on the Quality of English Language

- A few typos (e.g. lines 298, 301 etc.) are present but the overall quality of English is good.

Author Response

1. " It is a quantitative study, but the hypotheses as well as the hypothesis tests are missing." Thank you, clarification was needed and we have rewritten. We have provided the hypothesis in the introduction and the appropriate ways to report statistical tests (t-tests and ANOVA). Also (point 3 the typos have been corrected, thank you).

2. Questionable use of a picture

This is a tricky bit. We provided the picture (with young persons in the study, one of which was smiling - a picture showing ethnic diversity, which is good) to give the reader a sense of the intervention. We have substituted this picture with one where no faces can be seen. We completely agree with the reviewer that this (whether or how to include visual ethnoghraphy) is a very complex issue. Consider the below:

This is indeed an important aspect, and we agree it is worth considering carefully . Not showing a picture could also be argued to be questionable. Thank you for bringing it up. Figure 3 has now been changed (although we liked the other picture with one young man smiling).

All participants in the study have given their written consent for us to use their pictures in relation to the research. We find that one (or even two) pictures add to the reader’s grasp of the research setting. This is not the place to discuss this, but pictures in (Visual Anthropology and Education) are being reconsidered and re-introduced given that it gives participants real ownership of the research they take part of. Pseudo-anonymisation is becoming a whole discussion in research. Researchers have a duty to respect the dignity and rights of participants, ensuring that the use of images does not exploit or harm them. This includes being sensitive to cultural and personal boundaries regarding image use and considering the potential long-term consequences of disseminating (or not disseminating) visual data (Pauwels, 2008)

Reference:

Pauwels, L. (2008). Taking and using: Ethical issues of photographs for research purposes. Visual Communication Quarterly, 15(4), 243-257. https://doi.org/10.1080/15551390802415071

Reviewer 3 Report

Comments and Suggestions for Authors

General impressions
• Structure: The article is overall well-structured and it respects the section requirements. It needs slight modifications, better splitting of paragraphs, but otherwise no major changes are needed.
• Length: The length of the article is appropriate.

• Readability: The sentences are well structured, and there is an adequate flow of ideas.

• Relevance and novelty: Only few reports present similar cases, and this case itself presents an added value since it is rare for the transfer of learning between CT and CM so it is important to publish such a case and remind the scientific community of this possibility.

• Conclusions: The conclusion is well written and appropriate.

Major comments
The Introduction section presents the general idea; however, it lacks in updated references. I would recommend authors to carefully read the bibliography to find similar studies.
In the Assessment subsection please provide the Cronbach's alpha measure regarding the tests to determine reliability.

Minor comments
• Page 10 line 301 add one space character after the term CT5

Author Response

1. Structure: "The article is overall well-structured and it respects the section requirements. It needs slight modifications, better splitting of paragraphs, but otherwise no major changes are needed." Thank you, we have as can be seen from track changes ,rewritten the structure, slight and more modifications overall. 
• Length: "The length of the article is appropriate." We have deleted unnecessary sentences and added some for added clarity on the intervention and connection and differences between the two phenomena (CT and CM).
• Readability: "The sentences are well structured, and there is an adequate flow of ideas." Thank you.

• Relevance and novelty: "Only few reports present similar cases, and this case itself presents an added value since it is rare for the transfer of learning between CT and CM so it is important to publish such a case and remind the scientific community of this possibility." Thank you. We added some references on CT and CM in computer education.

• Conclusions: The conclusion is well written and appropriate.

Major comments
"The Introduction section presents the general idea; however, it lacks in updated references. I would recommend authors to carefully read the bibliography to find similar studies." Thank you - good point and we did now. Several new references were cited on transfer of learning (in context of computing education) and others on the discussed phenomena are now included in the manuscript and added to the reference list as can be seen in track change document.
"In the Assessment subsection please provide the Cronbach's alpha measure regarding the tests to determine reliability." Thank you for spotting this. The Cronbach’s alpha value is now provided in relation to the data analysis (section 3.7).

"Minor comments
• Page 10 line 301 add one space character after the term CT5" All corrected, thank you.

Round 2

Reviewer 1 Report

Comments and Suggestions for Authors

Review

First, I must thank the authors for providing more details about their intervention. Although they have done a lot of work in real classroom settings and their results must be interesting in the context of computing education, I have a series of questions about the study’s structure and choices, such as

Please correct me if I misunderstood something, but do you compare the development of CT and CM skills between two groups that are taught math and social sciences curriculum following 2 different instructional approaches, computing activities, and conventional teaching? How do you define conventional teaching, and how is it related to computing curriculum and skills? If the comparison group has not practiced CT and CM skills, how do we expect them to acquire them? Please explain.

If the case is to enrich conventional teaching with CT and CM skills, why don’t you additionally compare student learning in math and social sciences?

The researchers are not to follow the previous questions but to think about the study and provide the necessary information. I highly recommend that you reconsider the study’s details, provide more explanations if needed, and remove content that cannot be explained or confuses things...

I will write some specific comments for the paper.

-            Abstract: I am afraid that this abstract is too big. I really miss the study’s context in the first sentence of the abstract section and the study’s innovation in the second. Please reduce the size of the abstract, giving the most important information: what you did, what are the most important results, positive and negative, and why they are important

-            Introduction: “The research question was addressed through an intervention study, detailed in the Methods and Results sections.”: Please remember that in this section, you must explain how you will answer the research question!

“Before presenting the results, we will define CT and CM in the context of transfer of learning and discuss methods for measuring these two skills.”: I recommend the authors add a small paragraph to explain the following sections and their content.

Please reconsider the introduction section and compare it to your conclusions. In the introduction section, the authors say “the findings aim to inspire further research in computing education and provide guidance for high school aiming to prepare students for future  educational and career demands that require computational skills.“ Therefore, according to the research results, what does this study suggest?

-            Related work: Consider adding a small paragraph to identify the study’s choices based on the previously described literature.

-            Research design: This subsection should give the overall “image” of this study, meaning what is the purpose of the study and how this will be achieved. This means that the authors must explain the two study groups, how they were taught, and why. Please check the text.

-            Participants: Please explain how students were divided into classes. Was it a random distribution? Also, move here all the information about the study groups and their teachers (not their preparation).

-            Please use a subsection material to explain what the two groups were taught and what were the differences

-            Please use a subsection measuring instruments to provide all the information about the assessment tests. Please explain how you are going to assess the Transfer of Learning. 

-            Please consider using a subsection procedure to describe the steps (including teacher preparation)

-            Table 4: where is Figure 1?

Author Response

1. "do you compare the development of CT and CM skills between two groups that are taught math and social sciences curriculum following 2 different instructional approaches, computing activities, and conventional teaching? How do you define conventional teaching, and how is it related to computing curriculum and skills? If the comparison group has not practiced CT and CM skills, how do we expect them to acquire them? Please explain."

Us: 

Thank you and we have now clarified that the study's focus was to compare the impact of an enriched curriculum that integrates CT and CM skills into exisiting subjects (Math and Social Sciences) versus a a passive control (teaching sessions without such enrichment). We took great care to use the term “conventional” or “traditional” teaching (as these are normative terms) and whenever it was meaningful we wrote existing teaching methods. More importantly we defined what was meant by this type of teaching as well as the intervention.

Additionally, we explained that the comparison group received conventional instruction without specific CT and CM activities, allowing us to measure the impact of these additional skills when explicitly taught.

Now the question of design – why use a passive control group in an educational study? Is it not obvious that the intervention group (the class taught CT and CM) learns these skills whereas the control does not?

Using a passive control group in an educational intervention study, particularly when investigating Computational Thinking (CT) and Computational Modeling (CM) skills, was a strategic choice – also a pragmatic one given that it was a quasi-experimental tsudy. The passive control gorup establishes a baseline for comparison, allowing the measurement of relative effectiveness of the intervention against conventional teaching methods. This is elaborated on p. 5 (“The quasi-experimental design, with its passive control group, is well-suited for this investigation”)

2. "If the case is to enrich conventional teaching with CT and CM skills, why don’t you additionally compare student learning in math and social sciences?"

Us: We acknowledge the reviewer's point and  haverevised the manuscript to explain that the study’s primary goal was to assess the transferability of CT and CM skills when embedded in conventional subjects, rather than comparing overall subject proficiency.

3. "Abstract: I am afraid that this abstract is too big..."

Us: True again and we have revised to make the abstract more concise.

4. "Introduction: “The research question was addressed through an intervention study, detailed in the Methods and Results sections.”: Please remember that in this section, you must explain how you will answer the research question!"

Us: 

We thoroughly reviewed the manuscript (see the edition with red text) to remove any ambiguous content. Additionally, we provided clearer explanations of our study design, methodology, and the rationale behind our choices to improve clarity and transparency. 

We revised the introduction to include a brief overview of the methodology and approach used to answer the research question. Additionally, we added a small paragraph explaining the structure of the paper and what each section would cover.

5. "Please reconsider the introduction section and compare it to your conclusions. In the introduction section, the authors say “the findings aim to inspire further research in computing education and provide guidance for high school aiming to prepare students for future  educational and career demands that require computational skills.“ Therefore, according to the research results, what does this study suggest?"

Us: We revised the introduction to  align with the conclusions. Specifically, we now emphasize how the findings support the integration of CT and CM skills into high school curricula to prepare students for future educational and career demands. And, we discuss how learning transfer might be assessed.

6. "Related work: Consider adding a small paragraph to identify the study’s choices based on the previously described literature."

See last section/paragraph Background p.4

7. "Research design: This subsection should give the overall “image” of this study, meaning what is the purpose of the study and how this will be achieved. This means that the authors must explain the two study groups, how they were taught, and why. Please check the text.#

See: Study design (in Methods section)

8. "Participants: Please explain how students were divided into classes. Was it a random distribution? Also, move here all the information about the study groups and their teachers (not their preparation)."

Please see Participants (in Methods section)

9. "Please use a subsection material to explain what the two groups were taught and what were the differences"

Please see in Intervention sectios - "Material"

10. "Please use a subsection measuring instruments to provide all the information about the assessment tests. Please explain how you are going to assess the Transfer of Learning. "

Measurement Instruments (in Methods)

11. "Please consider using a subsection procedure to describe the steps (including teacher preparation)"

We have explained the intervention and the two days in subsection to Methods (called Teaching the Teachers p.8)

12. "Table 4: where is Figure 1?"

Thank you again - this is corrected and specified in Table 4

Reviewer 3 Report

Comments and Suggestions for Authors

The authors have made substantial revisions regarding the first sections of the paper. The paper has good flow and also good structure.

This paper should be published.

Author Response

"The authors have made substantial revisions regarding the first sections of the paper. "

We thank the reviewer and we have made further revisions to ensure an even stronger argument about the choice of design, intervention, findings.

"The paper has good flow and also good structure." Thank you.

Round 3

Reviewer 1 Report

Comments and Suggestions for Authors

I would like to thank the authors for their work on revising the paper and for their clear explanations. While I personally would not have chosen to compare a group that does not perform computational activities with one that does, the authors have provided a satisfactory explanation of their study, which is of great interest to the computing education community and beyond.

I have just one final comment:

Please remove lines 56-57 on the second page: “CT focuses on the thought processes students use when problem-solving with computers, while CM involves creating abstractions to understand and predict subject-specific phenomena using a computer.” This paragraph explains the structure of the paper and should not include definitions.

Author Response

"I would like to thank the authors for their work on revising the paper and for their clear explanations. While I personally would not have chosen to compare a group that does not perform computational activities with one that does, the authors have provided a satisfactory explanation of their study, which is of great interest to the computing education community and beyond."

Thank you for the constructive reviewer comments to earlier draft. Yes, we agree, the question about intervention-control studies versus say design based (or pre-post i.e. within subject design) is a vexed question. There is hardly one research design vastly superior to others.

"I have just one final comment: Please remove lines 56-57 on the second page: “CT focuses on the thought processes students use when problem-solving with computers, while CM involves creating abstractions to understand and predict subject-specific phenomena using a computer.” This paragraph explains the structure of the paper and should not include definitions."

True, we have done this now. Furthermore, there were some typos (like the sentence breaking off at the end of the intro about our research question - an unfinished sentence that we have removed as it was a typo.).